# Sleep and Mental Health Disturbances Due to Social Isolation during the COVID-19 Pandemic in Mexico

**DOI:** 10.3390/ijerph18062804

**Published:** 2021-03-10

**Authors:** Guadalupe Terán-Pérez, Angelica Portillo-Vásquez, Yoaly Arana-Lechuga, Oscar Sánchez-Escandón, Roberto Mercadillo-Caballero, Rosa Obdulia González-Robles, Javier Velázquez-Moctezuma

**Affiliations:** 1Centro de Sueño y Neurociencias, Benito Juárez, 06700 Mexico City, Mexico; gjovannat@gmail.com (G.T.-P.); ampvs1986@gmail.com (A.P.-V.); yoalysoph@hotmail.com (Y.A.-L.); 2Sleep Disorders Clinic, Universidad Autonoma Metropolitana, Iztapalapa, 09340 Mexico City, Mexico; lulygo1749@gmail.com; 3Neurophysiology Service, ABC Hospital, 05330 Mexico City, Mexico; oscarse@att.net.mx; 4Consejo Nacional de Ciencia y Tecnologia, Universidad Autonoma Metropolitana-Iztapalapa, 09340 Mexico City, Mexico; emmanuele.mercadillo@gmail.com

**Keywords:** COVID-19, sleep disturbances, anxiety, depression

## Abstract

The coronavirus disease (COVID-19) that broke out in China in December 2019 rapidly became a worldwide pandemic. In Mexico, the conditions requiring the declaration of a sanitary emergency were reached by the last week of March 2020, and health authorities’ limited mobility and imposed social isolation were the main strategies to keep the virus from spreading. Thus, daily living conditions changed drastically in a few days, generating a stressful situation characterized by an almost complete lack of mobility, social isolation, and forced full-time interactions with family members. Soon, complaints of sleep disturbances, anxiety, and symptoms of depression were reported. The present study reports the results of an online survey performed during the first two months of isolation. Questionnaires exploring sleep disturbances, anxiety, and depression were sent to people who responded to an open invitation. A total of 1230 participants filled out the sleep questionnaire, 812 responded to the anxiety questionnaire, and 814 responded to the depression questionnaire. Both men and women reported poor sleep quality, but women showed a higher proportion (79%) than men (60%); young women were more likely to be affected by social isolation. Concerning anxiety and depression, both sexes reported high similar symptoms. These data suggest that stressful conditions related to social isolation and the economic uncertainty caused by the pandemic may induce mental health disturbances, which may become worse with sleep restriction.

## 1. Introduction

Despite the antecedent of the bubonic plague in the Middle Ages or the Spanish Influenza in the early years of the 20th century, for the first time in history, humankind was able to witness the worldwide spread of a new disease in real-time. The coronavirus disease 2019 (COVID-19) broke out in Wuhan, China, in December 2019. The virus showed a high infection rate, fast transmission, and considerable mortality. As no vaccine or pharmacological treatment was available, authorities throughout the world considered it a serious threat to the population. After the disease reached Asia it spread to Europe and the Americas. Governments declared a sanitary emergency in almost all countries with a restriction on mobility and social isolation as common strategies to avoid or delay the spread of the disease.

In Mexico, the conditions needed to declare the sanitary emergency were reached in the last week of March 2020. By this time, most of the population was familiar with the high levels of people infected or deceased in other countries, so the sanitary indications were neatly followed. Mobility decreased significantly, while the social isolation conditions reflected an extreme concern of the population regarding health and safety. Thus, within a short time, a large number of families began to experience stressful conditions in their daily lives. The lack of social interactions, restrictions on human mobility, imposed full-time interactions with family members, and the new conditions of working from home or homeschooling may have influenced the modification of circadian rhythms (sleep–wake cycles). In addition, the alarming daily news and the uncertainty of the new economic conditions induced fear and agitation in big sectors of the population. Brought together, all of these conditions could have induced sleep disturbances and the onset of mental health impairments such as anxiety or depression.

Previous reports indicated that the 2003 pandemic of severe acute respiratory syndrome induced severe mental health disturbances, mainly among Asian populations [1]. As most of the current health efforts have been oriented towards medical attention for those infected and health workers, the psychosocial effects on the general population have been overlooked [2]. In a recent review of the psychological impacts of quarantine, some authors reported negative psychological effects [3]. In addition, recent reports from China indicated the presence of sleep disturbances, generalized anxiety disorders, and symptoms of depression in the population, all associated with quarantine conditions [4].

It is well known that isolation facilitates the onset of psychopathological manifestations, including anxiety, depression, sleep disturbances, hallucinations, and suicidal behavior [5]. Recently, the trigger of psychopathological manifestations was identified and includes boredom, stress, and sleep restriction [6]. Furthermore, it is well known that sleep and mental health disturbances present reciprocal causation [7]. Sleep disturbances often generate a negative vicious circle, worsening the quality of life of patients with mental health impairments. In addition, stressful situations such as social isolation often generate sleep disturbances that negatively influence mental health balance, mainly worsening the state of anxiety or depression [8,9,10].

In the last decades, a great volume of scientific papers has reported the negative effects of sleep restriction (for a review see [11]). These effects are observed both in the short and long term [12,13]. The short-term effects of sleep restriction are characterized mainly by an impairment of executive functions, including verbal and non-verbal working memory, attention deficits, and emotional and motivational regulation, as well as difficulties in problem-solving, among others (for a review see [14]). The negative effects of sleep restrictions can be observed in a broad range of ages and are particularly critical in children’s day-time functioning [15]. Furthermore, mood alterations due to sleep restriction also have been repeatedly reported [16]. These mood changes are particularly in stressful working situations, such as in hospitals. The effects of sleep restriction on the increase of conflicts within groups have been repeatedly reported [17]. Medical residents display increasing impairments in their ability to show empathy towards patients and towards other members of the group [18]. Thus, stressful conditions within the family group due to the sudden lockdown have an impact on the circadian rhythms and, as consequence, on sleep, resulting in a significant level of sleep restriction.

As shown in preceding paragraphs, quarantine and isolation conditions to safeguard human beings from COVID-19 could produce stressful circumstances that cause negative mental health effects, such as depression, anxiety, and sleep disturbances [3,4,5]. An additional factor that could be presented during quarantine is the increase of exposure to light stimulation at night by using bright screens, such as cell phones, TVs, or tablets, which may suppress the release of melatonin, resulting in major disturbances of circadian rhythms and sleep disorders [19].

Knowing the effects of quarantine on sleep and mental health of the world’s population, it is essential to design psychological and social interventions during and after the current pandemic. In the case of Mexico, results that could guide such interventions and contribute to national and international scientific knowledge have not yet been reported. Thus, the purpose of the present study was to explore the presence of sleep and/or mental health disturbances, as well as, the use of bright screens at night in the Mexican population while living in quarantine conditions to prevent COVID-19 infections.

We hypothesized that adult Mexican people living in quarantine to prevent COVID-19 infection manifest sleep disturbances and symptoms of mental disorders such as anxiety and depression.

## 2. Materials and Methods

### 2.1. Participant Recruitment

An open invitation was sent via Internet to participate in the present survey. Google forms focused on questionnaires about sleep habits, sleep quality, and mood were sent. We used our social networks and those of the researchers and specialists affiliated with our group to invite people who voluntarily wanted to fill out the forms. Subjects included in the study and those who filled out the questionnaires were older than 18 years. Incomplete or incoherent responses were discarded. In return, participants received the results of their questionnaires accompanied by particular recommendations for improving their sleep quality, as well as suggestions for sleep hygiene and relaxing anti-stress indications to improve their mood and decrease their anxiety levels. The protocol followed the principles outlined in the Declaration of Helsinki and it was reviewed and approved by the Institutional Review Board. All the participants provided their written informed consent to participate in this study and their data were protected and kept in strict privacy.

### 2.2. Electronic Instruments

Questionnaires were electronically available from 28 March to 26 May 2020. Although social isolation officially started on 24 March, several families, particularly in the big cities of Mexico, decided to decrease their mobility two or three weeks before. Thus, the period we defined allowed us to obtain data regarding the initial week of isolation. Participants filled out demographic information (age, sex, profession, and place of residence), a sleep diary for the final week, the Pittsburgh Sleep Quality Index (PSQI), the Generalized Anxiety Disorder 7-item Scale (GAD), and the Patient Health Questionnaire (PHQ-9). Filling out these questionnaires took about 10 min.

The PSQI includes 19 items that assess sleep schedules, sleep latency, frequency, and the duration of awakening from sleep and nocturnal signs that suggest sleep disorders. The version we used was the one validated and already used among Mexican samples, which considers a set point >5 to indicate low sleep quality [20].

The GAD scale explores the presence of symptoms representing the generalized anxiety diagnosis according to the Diagnostic and Statistical Manual of Mental Disorders, 4th Ed. (DSM-IV). It includes 7 items concerning anxiety data in the two weeks before when the participant answers. We used the Spanish version [21] previously validated among Mexican samples [22] to identify the presence of mild, moderate, or severe anxiety.

The PHQ-9 questionnaire assesses the presence of depressive symptoms during the 2 weeks before the test, according to the DSM-IV criteria. With 9 specific items, the Mexican validated version [23] indicates no signs, depression that could require treatment, and depression that needs pharmacological or psychological therapy.

As bright light exposure can alter circadian rhythms and induce sleep restriction, the daily hours of exposure to electronic screens or devices were analyzed. To this aim, a special questionnaire was designed to explore how many hours the subjects were exposed daily to a bright screen as well as the moment in which this exposure took place. Additionally, in the same questionnaire we explored the activities the subjects performed before going to bed.

### 2.3. Data Analyses

Databases were generated, and statistical analyses were performed using the Statistical Package for Social Sciences (SPSS) 21st version. The chi-square test was used to analyze proportions, and one-way ANOVA with the Duncan test, Binomial exact test, and t-test for independents variables were used for group comparisons. A *p*-value of 0.05 was established as a maximum to consider significant differences.

## 3. Results

A total of 1230 questionnaires were received from people living in 31 of the 32 federal regions in Mexico; most of them (62.4%) came from the central region, most of the participants were women (864, 70.2%), and most of them were around 40 years old or younger (713, 58%).

Concerning sleep quality to explore sleep disturbances, Figure 1 summarizes the results about the PSQI items. A high percentage of the sample reported a sleep latency of more than 15 min (A), sleep duration for most of the participants was less than 7 h (B), subjective sleep quality reported by the participants showed no major differences between good/very good compared with bad/very bad (C), sleep efficiency was mainly reported as less than 85% (D), a high percentage of presence of any signs of sleep disorders and a very low percentage of the absence of sleep disorders was observed (E), and more than 80% of the participants reported daytime dysfunctions (F).

Table 1 summarizes the statistical analysis by using the chi-square test for the items included in the PSQI. Sleep latency was different between sexes, with women showing longer sleep latencies than men. Sleep duration also differed between sexes; a higher percentage of women slept less than 5 h. Regarding sleep quality, a higher proportion of women (55%) reported bad and very bad sleep quality, whereas men under the same parameter showed a lower proportion (37.9%). Sleep efficiency showed a similar trend and differences were represented in extreme values. The percentage of women with a sleep efficiency of 85 or more was lower compared to men. On the contrary, the percentage of women with sleep efficiency below a value of 65 was higher than the percentage of men. The presence of sleep disorders was also different between sexes since a high proportion of women experienced these problems once or twice a week. Concerning the use of sleep medication, no significant differences were observed between sexes. Differences between sexes were observed regarding daytime dysfunctions, with a higher percentage of women affected.

Table 2 shows the results for the chi-square test when participants were grouped by age: young = 18–40 years and old ≥ 40 years old. The younger group showed longer sleep latencies, a higher proportion of young participants reported a sleep duration of fewer than 5 h, more than a half of the younger participants reported bad or very bad sleep quality, a higher proportion of young participants reported low sleep efficiency <65%, younger participants reported a higher frequency of sleep disorders once or twice a week, the older group reported a high percentage of participants using drugs several times, and daytime dysfunctions showed a proportion of more than half of younger participants reporting this issue as moderate or severe.

Figure 2 summarizes the overall results concerning the PSQI evaluation of sleep quality grouped by ages. The ≤40-year-old women seem to be the most affected group and showed differences from the other age groups. Similarly, ≤40-year-old men reported more sleep disturbances than the >40-year-old participants. No significant differences were observed between men and women who were older than 40.

Regarding anxiety, 34.8% of women reported mild symptoms, 18.3% moderate, and 18% severe. In the case of men, 38.1% reported mild symptoms, 16.9% moderate, and 19% severe. No differences between genders or ages concerning anxiety were identified.

Regarding depression, results for the PHQ-9 indicated no differences between genders or age, with *p* = 0.664. However, 24.5% of women and 18.6% of men reported moderately severe and severe depression symptoms. In addition, the results show that younger females reported a higher percentage of depression symptoms (26.4%) than older males (14.7%)

Furthermore, when the use of electronic devices (TV, cellphone, or any kind of computer) was analyzed, the results indicate that more than half of the sample (68.9%) was exposed to bright lights for at least 5 h daily. Women reported a higher percentage of screen use at around 10 h daily (17.4%) and reported a longer used of screens compared to men (13.4%) *p* ≤ 0.01. When activities related to preparing for bed were explored, a high percentage of the participants (62.3%) reported the use of some kind of electronic device as part of their activities; prior to sleep no differences were observed regarding age.

## 4. Discussion

The purpose of this study was to explore the presence of sleep and/or mental health disturbances in the Mexican population while living under quarantine conditions to prevent COVID-19 infections. Our results included the evaluation of the first eight weeks of social isolation in Mexico and focused mostly on big cities. Similar to an Italian report, a high percentage of females responded to the survey (more than 70% in both studies) [24,25], which suggests that women manifest a greater concern for their own and their family’s issues.

According to our results, a high percentage of the participants’ sleep was severely affected, with reports of poor quality of sleep. Furthermore, the percentage of women affected was higher than men. Additionally, younger people seemed to be more sensitive to conditions of isolation and their sleep was more affected than that of older people. Women under 40 proved to be the most sensitive population to sleep disturbances. These results are similar to those reported in early studies on Italian [25] and Chinese samples (4).

These results can be explained by the role that young women usually play within the family home. In addition, the increase in domestic violence, mainly affecting young women, has been recently reported [26].

These results may reflect important sleep restriction conditions presented among the Mexican population while facing quarantine conditions. Thus, the present data should be taken as an extreme concern since current knowledge about sleep restriction suggests deficits in behavioral performance and mood swings [27,28], and research conducted on medical students with sleep restriction reported an impairment of empathy and an increase of in-group conflicts [29]. In addition, similar to what was reported in other countries during quarantine, the calls to 911 reporting domestic violence and violence against health workers in Mexico have considerably increased [30,31].

Moreover, and as reflected in our results, disturbances of the circadian rhythm play a role in the onset of sleep disturbances. In this sense, the purpose of the study included the exploration of the use of bright screens at night. Participants declared a frequent use of these instruments with a clear trend of using them at night. As has been reported, the excess of blue light stimulation provided by electronic devices interferes with the onset of sleep [32]; particularly in school children and adolescents, the excessive use of electronic devices induces a delayed bedtime, shortened sleep duration, and longer sleep latency [33,34]. Thus, it seems that quarantine conditions facilitate the use of TV and electronic devices among the entire family, which may affect circadian rhythms, resulting in a cascade of adverse events that includes sleep restriction and reduced sleep quality [19]. Moreover, the association between the use of electronic devices, sleep disturbances, and depression has been confirmed for adolescents. Thus, the presence of depressive symptoms observed in our sample can be due, at least partially, to sleep disturbances induced by the excessive use of electronic devices [35], since there is a clear relationship between young women using electronic devices for 10 h daily and sleep disturbances.

Concerning ages, Yu et al. reported in 2018 [36] that social isolation was the main factor for sleep disturbances in older people; when comparing their results obtained from 639 participants in Taiwan, the authors indicated that social isolation and not loneliness was the main factor for sleep disturbances. Controversially, our results do not confirm this report. Rather, we observed that younger people manifested sleep disturbances included in the PSQI, whereas older people showed only slight changes in sleep. Furthermore, young women appeared to be more vulnerable to lockdown conditions, showing an increased frequency of sleep and mental health disturbances. It is possible that the role and responsibilities that young women have within Mexican families impose additional pressure on their mental stability.

It remains to be elucidated why young women seems to be more vulnerable to sleep loss in lockdown conditions. Some studies reported sleep disturbances in pregnancy and during premenstrual syndrome. However, to date, endocrine factors cannot fully explain the increased frequency of sleep and mental disturbances in young women.

Furthermore, social isolation induces a decrease in solar light stimulation that negatively affects both circadian rhythm synchronization and mood [37].

Mexico City is the most crowded region of the country with nearly 20 million people, and its authorities decided to implement measures of social isolation to avoid spreading the virus. Health authorities reported that more than 70% of the people decided to stay at home with minimal social interactions.

The present results are in accordance with other recent publications reporting mental health disturbances due to lockdown conditions all over the world [38]. Concerning generalized sleep restriction, public health strategies should be discussed to avoid future adverse health outcomes. Since 2015, the American Academy of Sleep Medicine and the Sleep Research Society addressed the relationships between sleep duration and the risk of suffering from a wide variety of diseases including diabetes, stroke, endocrine dysfunction, mental illness, and cancer. A panel of experts in sleep medicine reviewed the available scientific evidence in more than 5300 scientific articles. After more than 12 months of review and discussion a final statement was disclosed. The consensus statement indicated than sleeping less than 7 h regularly increases the risk of death. This death risk was associated with the presence of obesity, diabetes, heart diseases, and depression. In addition, there is also a greater risk of accidents [39]. Thus, there is an increased probability that this prolonged period of sleep restriction leads to an increase in the frequency of dangerous diseases among the general population.

Our exploratory study involved samples reflecting the first weeks of quarantine. The most significant results indicate that a high percentage of the participants reported poor sleep quality and that women and young people were more sensitive to such effects. In addition, quarantine conditions seemed to facilitate the use of electronic devices at night, which can affect the quality of sleep and favor the presence of depression, particularly in young people. These results are important for designing mental health intervention strategies during quarantine that include sleep hygiene habits and practices to maintain or improve the quality of sleep in the population, and to prevent the presence of symptoms of depression, with emphasis on women and youth.

The present results should be taken with caution because the population evaluated came only from the country’s large cities and not from other regions in which quarantine conditions and related information may vary. Furthermore, our sample was limited to participants who could access digital devices to answer the questionnaires, leaving out a large part of the population that does not have access to such resources. Importantly, we do not have previous measures on sleep quality, depression, anxiety, or the use of electronic devices in equivalent samples before quarantine conditions, so, although our results coincide with previous reports in Italy and China, they cannot be strictly associated with only quarantine conditions. Future studies can be conducted at different times in the pandemic as a comparative measure.

The pandemic and lockdown conditions have lasted more than expected, so new surveys should be done to explore the effects of several months of social isolation on sleep, anxiety, and depression, to understand in more detailed the effects of quarantine conditions on mental health.

## Figures and Tables

**Figure 1 ijerph-18-02804-f001:**
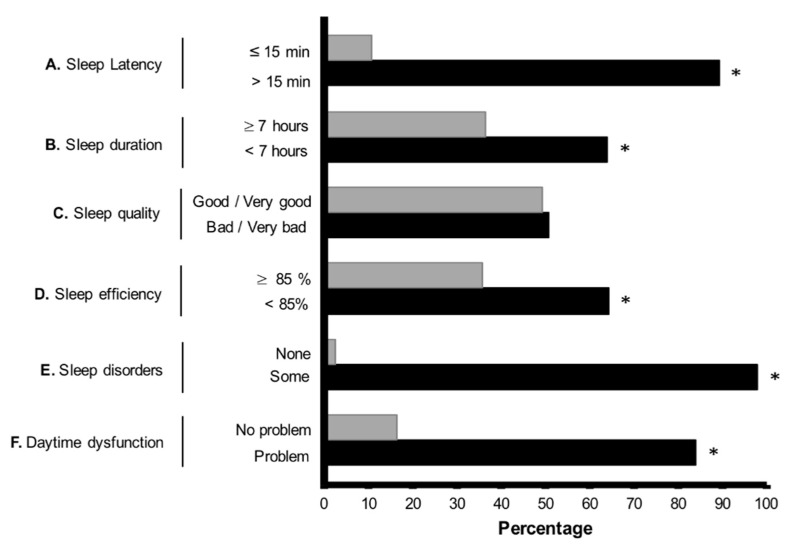
Percentage of responses given for the Pittsburgh Sleep Quality Index items. * Indicates significant differences at *p* < 0.001 when the exact binomial test was applied.

**Figure 2 ijerph-18-02804-f002:**
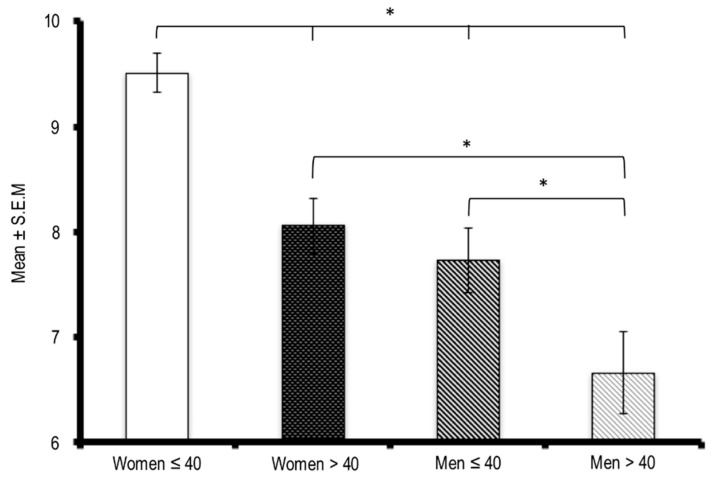
Mean and standard deviation representing the presence of sleep disturbances evaluated by the Pittsburgh Sleep Quality Index grouped by gender and age (≤40 y.o. and >40 y.o.). * Indicates differences at *p* < 0.05 when the ANOVA test and post hoc Duncan test were applied.

**Table 1 ijerph-18-02804-t001:** Gender differences for the Pittsburgh Sleep Quality Index items according to the chi-square test applied.

Gender	Pittsburgh Sleep Quaility Index Item				Chi-Square (n, *p*-Value)
Sleep latency					
	5–15 min	16–30 min	31–60 min	>60 min	n = 1154 *p* < 0.001
Women	8.3%	25.9%	32.6%	33.1%	
Men	17.9%	37.9%	25.2%	19.0%	
Sample %	10.7%	28.9%	30.8%	29.5%	
Sleep Duration					
	≥7 h	6 h	5 h	<5 h	n = 1154 *p* < 0.05
Women	35.4%	27.1%	22.3%	15.2%	
Men	38.6%	27.9%	24.8%	8.6%	
Sample %	36.2%	27.3%	23.0%	13.5%	
Sleep Quality					
	Very good	Good	Bad	Very bad	n = 1154 *p* < 0.001
Women	7.5%	37.5%	44.9%	10.1%	
Men	15.5%	46.6%	31.7%	6.2%	
Sample %	9.5%	39.8%	41.6%	9.1%	
Sleep Efficiency					
	≥85%	75 a 84%	65 a 74%	<65%	n = 1005 *p* < 0.001
Women	34.0%	19.7%	20.6%	25.6%	
Men	40.9%	26.6%	13.5%	18.9%	
Sample %	35.8%	21.5%	18.8%	23.9%	
Sleep Disorders					
	Not during the past month	Less than once a week	Once or twice a week	Three or more times	n = 1154 *p* < 0.001
Women	2.0%	64.2%	32.3%	1.5%	
Men	3.4%	77.6%	17.9%	1.0%	
Sample %	2.3%	67.6%	28.7%	1.4%	
Sleep Medications					
	Not during the past month	Less than one a week	Once or twice a week	Three or more times	n= 1154 *p* < 0.57
Women	78.6%	7.8%	5.9%	7.8%	
Men	82.4%	6.2%	4.8%	6.6%	
Sample %	79.5%	7.4%	5.6%	7.5%	
Daytime Dysfunction					
	No problem at all	Very slight problem	Somewhat of a problem	A very big problem	n = 976 *p* < 0.001
Women	13.9%	39.5%	42.4%	4.2%	
Men	22.7%	44.7%	29.4%	3.1%	
Sample %	16.2%	40.9%	39.0%	3.9%	

**Table 2 ijerph-18-02804-t002:** Differences for the Pittsburgh Sleep Quality Index items according to age grouping when the chi-square test was applied.

Age	Pittsburgh Sleep Quaility Index Item				Chi-Square (n, *p*-Value)
Sleep Latency					
	5–15 min	16–30 min	31–60 min	>60 min	n = 1157 *p* < 0.001
18–40	7.2%	24.7%	34.1%	34.1%	
>40	16.7%	35.6%	25.2%	22.5%	
Sample %	10.8%	28.9%	30.7%	29.6%	
Sleep Duration					
	≥7 h	6 h	5 h	<5 h	n = 1157 *p* < 0.001
18–40	34.2%	27.3%	21.5%	17.0%	
>40	39.4%	27.0%	25.2%	8.3%	
Sample %	36.2%	27.2%	22.9%	13.7%	
Sleep Quality					
	Very good	Good	Bad	Very bad	n = 1157 *p* < 0.001
18–40	4.2%	37.0%	48.0%	10.8%	
>40	18.2%	44.4%	31.1%	6.3%	
Sample %	9.6%	39.8%	41.5%	9.1%	
Sleep Efficiency					
	≥85%	75 a 84%	65 a 74%	<65%	n = 1007 *p* < 0.003
18–40	31.7%	22.1%	19.2%	26.9%	
>40	42.3%	20.6%	18.0%	19.1%	
Sample %	35.7%	21.5%	18.8%	23.9%	
Sleep Disorders					
	Not during the past month	Less than once a week	Once or twice a week	Three or more times	n = 1157 *p* < 0.001
18–40	1.3%	66.9%	31.0%	0.8%	
>40	4.1%	68.9%	24.8%	2.3%	
Sample %	2.3%	67.7%	28.6%	1.4%	
Sleep Medications					
	Not during the past month	Less than one a week	Once or twice a week	Three or more times	n= 1157 *p* < 0.003
18–40	82.5%	6.9%	5.3%	5.3%	
>40	74.8%	8.1%	6.3%	10.8%	
Sample %	79.5%	7.3%	5.7%	7.4%	
Daytime Dysfunction					
	No problem at all	Very slight problem	Somewhat of a problem	A very big problem	n = 978 *p* < 0.001
18–40	12.1%	36.9%	45.6%	5.4%	
>40	22.6%	46.9%	29.0%	1.5%	
Sample %	16.3%	40.9%	39.0%	3.9%	

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
