# Peer review of "Sleep and Mental Health Disturbances Due to Social Isolation during the COVID-19 Pandemic in Mexico"

_ijerph, 2021, doi:10.3390/ijerph18062804_

Round 1
Reviewer 1 Report
Please, have in mind the link between the objective of your study, approach, and results. The readers want to easily understand the link between Theoretical Background – Research Purpose – Data – Empirical Results – Discussion. Be as specific as possible and also as simple as possible.
Lack of research constraints and limitations.
The significance of the findings should be explained to the reader more clearly. Why is this study important? The research significance is not sufficient.
Line 95-97 Hypothesis is not well-established in theory.
Line 99-111 Please explain in more detail how you chose those respondents
Author Response
Enclosed you will find the detailed list of modifications done following the indications of the referees.

Reviewer 2 Report
In page 3, there seems to be something missing in the lines 134-135.
Explanation with regards to how the usage of electronic devices was measured lacks clarity in the methodology section.
It is interesting to note that the authors had mentioned the fact that Mexican women spend more time on supervision of scholarly activities during the lockdown (page 8, lines 231-232)...however, whether there was any measure of that in this study isn't mentioned and statistics to prove that would have added more weightage to the paper.
The reasons discussed for linking social isolation to sleep deprivation is quite superficial as there could be various other reasons for sleep deprivation apart from just social isolation. Were all the confounding factors accounted for in the data analyses? If yes, which confounding factors were included and why? This needs to be included in the methodology-data analyses section as well and the results discussed accordingly.
